# Effects of Interaction of Protein Hydrolysate and Arbuscular Mycorrhizal Fungi Effects on Citrus Growth and Expressions of Stress-Responsive Genes (*Aquaporins* and *SOSs*) under Salt Stress

**DOI:** 10.3390/jof9100983

**Published:** 2023-09-29

**Authors:** Qi Lu, Longfei Jin, Peng Wang, Feng Liu, Bei Huang, Mingxia Wen, Shaohui Wu

**Affiliations:** 1Zhejiang Citrus Research Institute, Taizhou 318026, China; lu13600623887@163.com (Q.L.); jlf_0511@163.com (L.J.); lfz5799@163.com (F.L.); zj_citrus_hb@163.com (B.H.); wenmx198@126.com (M.W.); liuxingwsh@163.com (S.W.); 2College of Horticulture and Gardening, Yangtze University, Jingzhou 434025, China

**Keywords:** citrus, salt, biostimulants, protein hydrolysate, AMF, *Aquaporins*, *SOSs*

## Abstract

Protein hydrolysates (PHs) and arbuscular mycorrhizal fungi (AMF) are environmentally friendly biostimulants that effectively promote crop growth and alleviate the damage from abiotic stress. However, the physiological and molecular regulatory mechanisms are still unclear. This study aimed to explore the effects of PHs and AMF on growth, mineral nutrient absorption, and expression of *Aquaporins* and *SOSs* in Goutoucheng (*Citrus aurantium*) under salt stress. Results showed that PH application and AMF inoculation significantly promoted plant growth and enhanced mineral element absorption and sodium effluxion in citrus under salt stress. The biomass, root activity, leaves mineral nutrition contents in PHs, AMF, and combined (PHs and AMF) treatments were significantly higher than those of control. Leaves sodium content in three treatments was significantly lower than in the control. AMF and combined treatments showed dominant effects than PHs alone. Besides, PHs interacted with AMF on growth, nutrient absorption, and sodium effluxion. Importantly, AMF and PHs induced stress-responsive genes. *PIP1*, *PIP3*, *SOS1*, and *SOS3* expression in PHs and AMF treatments was significantly higher than control. Thus, it was concluded that AMF and PHs enhanced the salt tolerance of citrus by promoting nutrient absorption and sodium effluxion via up-regulating the expression of *PIPs* and *SOSs*. The mixed application of PHs and AMF had a better effect.

## 1. Introduction

Citrus is one of the most important economic fruit crops worldwide, with a global annual production of nearly 119 million tons in 2021 (FAO: https://www.fao.org/faostat/zh/#home. Accessed on 12 September 2023). Notably, facility cultivation has been widely applied in citrus to produce high-quality fruit [1]. However, a lack of rain, high temperature, significant evaporation, and excessive application of fertilizers tend to cause salt accumulation and soil secondary salinization [2]. Salt stress in greenhouses is commonly caused by soil secondary salinization [3]. Importantly, citrus is susceptible to salt stress [4]. Under salt stress, citrus leaf yellowing, leaf tip scorch, leaf fall, and even tree death often occur, which seriously affects the stability of facility citrus yield and quality, as well as the improvement of planting efficiency [5]. Additionally, excessive sodium ions can cause plant ionic toxicity [6]. The salt overly sensitive (SOS) signal transduction pathway plays an essential regulatory role in sodium ion homeostasis in plants [7,8,9]. Furthermore, too much salt in the soil reduces soil water potential, inhibiting plant roots from absorbing water [10]. Aquaporins (AQPs), especially plasma membrane intrinsic protein genes (*PIPs*) and tonoplast intrinsic protein genes (*TIPs*), play critical roles in regulating water uptake by crop roots [11]. Moreover, overexpression of AQP proteins imparts salt tolerance in transgenic plants through enhancing water acquisition [12,13].

Protein hydrolysates (PHs) are a mixture of oligopeptides, polypeptides, and amino acids formed by the hydrolysis of plant proteins [14]. Numerous investigations suggest that PHs promote plant growth [15], improve yield and quality [14], and enhance crop resistance to abiotic stress, such as salinity [16], through activating several molecular and physiological biological processes by foliar or root applications of plant-derived PHs. For example, seed priming with PHs improves *Arabidopsis* growth and tolerance of abiotic stresses by reducing flavonoids, terpenoids, and some degradation/conjugation compounds of cytokinin, auxin, and gibberellin [17]. Under salt stress, PH applications enhanced the salt tolerance of lettuces and tomatoes by regulating the metabolism of auxin and ethylene [18]. However, studies on PH application performance and action mechanisms are rarely reported in citrus cultivation.

Arbuscular mycorrhizal fungi (AMF) extensively established a symbiotic relationship with host plants in salt soil [19]. Additionally, inoculation with AMF strains isolated from salt soil played a significant role in improving the salt tolerance of crops [20]. Glomalin-related soil protein (GRSP) is a unique glycoprotein secreted by hyphae and spore walls of AMF, significantly contributing to soil particle aggregation and carbon sequestration [21]. Studies have shown that GRSP regulates plant response to abiotic stress [22,23]. Moreover, AMF regulated the expression of stress-related genes in plants and enhanced the host’s abiotic stress tolerance. For example, AMF induced the expression of *AQPs* and *SOSs* in tomatoes, enhancing the effluxion of sodium ions and water absorption under salt stress, thus enhancing salt tolerance [24,25]. In pistachio, AMF stimulated nutrient uptake and maintained ionic homeostasis through up-regulating *SOS1* and *PIP2.4* expression [26]. Since citrus has few root hairs, the mycorrhizal structure formation from AMF colonizing roots can improve water absorption and nutrients through the inner and external mycelium [27]. Numerous studies have also reported that artificial inoculation of AMF enhanced fruit quality [28] and tolerance to various abiotic stresses [29,30]. For citrus, AMF mainly induces ion balance, photo-chemistry, and an antioxidant defense system in response to salt stress [31,32]. Studies have also shown that AMF induced the AQP gene in trifoliate orange under salt stress, which enhanced root water absorption [30]. However, the molecular mechanism of mycorrhizal on salt tolerance enhancement in citrus remains unclear.

Although using PHs or AMF alone can promote plant growth and improve plant resistance, whether the combined application of PHs and AMF will synergistically enhance the salt tolerance of citrus rootstocks is entirely unknown. Goutoucheng (*Citrus aurantium*) is a local citrus germplasm resistant to salt stress and is widely used as rootstock in coastal orchards [33]. This study analyzed the effects of PHs and AMF on Goutoucheng seeding growth, physiological activities, and the expression of stress-responsive genes (SOSs, TIPs, and PIPs) under salt stress. This study was expected to reveal the interaction mechanism between PHs and AMF in response to the salt tolerance of citrus plants.

## 2. Materials and Methods

### 2.1. Materials and Treatments

Goutoucheng seeds were collected from Zhejiang Citrus Research Institute, and the seedling propagation was conducted following by previous research [34]. A total of 4 two-month-old seedlings were planted in pots (8 L) containing autoclaved salinized soil (containing available nitrogen 90.08 mg/kg, available phosphorus 10.71 mg/kg, available potassium 121.50 mg/kg, organic matter 3.34%, soluble salt 0.63%, pH 6.07) from greenhouses of coastal facilities. The commercial legume-derived PHs were provided by Italpollina S.p.A (Verona, Italy). The compounds of protein hydrolysate contain aspartic acid (3.16%), threonine (1.15%), serine (1.35%), lysine (5.37%), glycine (1.21%), alanine (1.27%), cysteine (0.52%), valine (1.38%), methionine (0.43%), isoleucine (1.33%), leucine (2.23%), tyrosine (0.91%), and phenylalanine (1.50%). The arbuscular mycorrhizal fungus used in this study was *Rhizophagus intraradices* (Ri), which was isolated from the soil of a citrus orchard in southern China [35]. A 2 × 2 double factors experiment was conducted in this research. PHs treatment followed the product instructions: PHs solution (0.2%) was irrigated 10 days after seedling transplantation, once a week for 10 weeks. Ri treatment: 20 g Ri (1 g contains 20 spores) was inoculated in one pot at seedling transplantation. RP treatment: PHs and Ri were applied together. CK: distilled water instead of PHs irrigation and non-inoculation Ri were conducted.

### 2.2. Sample Collection and Physiological Index Determination

The samples were collected after 16 weeks of growth. The plant height, stem diameter, and leaf number were directly determined. Shoot dry weight was detected after drying in a hot air-circulating oven. The leaf nitrogen (N) content was assessed using an Automatic Kjeldahl Azotometer (KT8200, FOSS, Genhagen, Sweden). The phosphorus (P), potassium (K), calcium (Ca), magnesium (Mg), and sodium (Na) content of leaves was detected using an inductively coupled plasma emission spectrometry (Vista-MPX, Varian, Palo Alto, CA, USA) [36]. The root structure was scanned using MICROTEK ScanMaker i800 plus (MICROTEK, Technology Co., Ltd., Xinzhu, China) and analyzed using SC-E software (Hangzhou Wanshen Detection Technology Co., Ltd., Hangzhou, China. http://www.wseen.com/ProductDetail.aspx?id=9&classid=29. Accessed on 29 September 2023). The root activity was measured using the TTC reduction method [37]. The 1–2 cm long root segments were collected and stained with trypan blue in a lactic acid phenol solution for mycorrhizal staining [25], and the microstructure was observed using an optical microscope. The colonization rate of mycorrhizal fungi in roots was estimated as the percentage of the root segment length colonized by mycorrhizal fungi versus the observed root segment length. Furthermore, the GRSP was extracted and measured according to previous research [38]. The samples for gene expression analysis were immediately frozen in liquid nitrogen and then stored at −80 °C for later use.

### 2.3. RNA Extraction and Gene Expression Analysis

Total RNA was extracted using an ultra-pure total RNA extraction kit (SIMGEN, Hangzhou, China). RNA concentration of different treatments ranged from 83 to 400 ng/μL. Following integrity and purity testing, 1 μg of the obtained RNA was reversely transcribed into cDNA. The first-strand cDNA was synthesized using PC18-TRUEscript 1st Strand cDNA Synthesis Kit (Aidladb, Beijing, China). qRT-PCR was performed using SYBR Green qPCR Mix (Aidladb, Beijing, China) in CFX96 Real-Time PCR Detection System (Bio-rad, Hercules, CA, USA). Detailed experimental procedures followed the kits’ instructions. The reaction program began with initial incubation at 94 °C for 2 min, followed by 40 cycles of 94 °C for 10 s and 60 °C for 30 s. The *TIPs*, *PIPs*, and *SOSs* gene sequences were downloaded from the citrus genome (http://citrus.hzau.edu.cn/index.php. Accessed on 29 September 2023). The primers were designed using Primer 3 software, and the primer sequences are listed in Appendix A. The 2^−ΔΔCT^ method was used to quantify relative normalized gene expression levels [39]. Actin gene was used as an internal reference.

### 2.4. Data Analysis

Citrus plant phenotypic assessment was performed in five independent trials with at least 15 plants per treatment. All data are expressed as mean ± standard error (SE). Sigmaplot 10.0 software was used for plotting. The significant differences in biomass, nutrient element content, and qRT-PCR results among four treatments were analyzed using the least significant difference (LSD) test at the 0.05 level. Furthermore, the interaction (S × P) between the two factors PHs and Ri was analyzed using the SAS 8.1 software (SAS Institute Inc., Cary, NC, USA).

## 3. Results

### 3.1. Effects of PH and Ri on Plant Growth of Citrus Exposed to Salt Stress

Under salt stress, PH application and Ri inoculation promoted the growth of citrus seedlings. The effect of PH application together with Ri inoculation was better than individual processing (Figure 1). Compared to CK, PH application significantly increased the plant height, stem diameter, leaf number, and shoot dry weight by 40.23%, 37.50%, 58.52%, and 77.05%, respectively. Additionally, Ri inoculation significantly increased the plant height, stem diameter, leaf number, and shoot dry weight by 131.03%, 81.25%, 158.82%, and 227.00%, respectively. Furthermore, PH application and Ri inoculation significantly increased the plant height, stem diameter, leaf number, and shoot dry weight by 174.71%, 75.00%, 169.12%, and 480.32%, respectively (Table 1). Thus, an interaction between factors (PHs × Ri) was observed for stem diameter and leaf number.

### 3.2. Effects of PHs and Ri on Leaves Mineral Element Content of Citrus Exposed to Salt Stress

Compared to CK, PH application, Ri inoculation, and PHs + Ri treatment significantly increased N, P, and Ca and decreased Na content in citrus leaves under salt stress. Additionally, the PH application significantly increased K and Mg content. Furthermore, Ri inoculation and PHs + Ri treatment significantly decreased K content. Thus, an interaction between factors (PHs *×* Ri) was observed for N, P, Ca, Mg, and Na contents (Table 2).

### 3.3. Effects of PHs and Ri on Root Structure of Citrus Exposed to Salt Stress

Under salt stress, PH application and Ri inoculation promoted root growth of citrus seedlings (Figure 2). PH application also significantly increased the lateral root number, total root length, and root activity by 21.17%, 16.39%, and 52.94%*,* respectively. Additionally, Ri inoculation significantly increased the lateral roots number, root volume, total root length, root projection area, root surface area, and root activity by 80.04%, 200.00%, 120.24%, 124.52%, 124.79%, and 94.12%*,* respectively. Moreover, PH application with Ri inoculation significantly increased lateral root number, root volume, total root length, root projection area, root surface area, and root activity by 119.63%, 250.00%, 149.34%, 143.22%, and 135.29%*,* respectively (Table 3).

### 3.4. Effects of PHs on Root Colonization of Citrus Inoculated by Ri under Salt Stress

Mycorrhizal fungal colonization was visible in the roots of Ri-inoculated plants (Figure 3), but no signs of mycorrhizal colonization were observed in non-inoculation citrus roots. Moreover, PH application and Ri inoculation significantly increased the hyphae colonization rate, total colonization rate, and hyphal density by 24.41%, 20.86%, and 28.29%*,* respectively (Table 4).

### 3.5. Effects of PHs and Ri on Rhizospheric GRSP Contents of Citrus Exposed to Salt Stress

Compared to CK, Ri inoculation significantly increased the contents of EE-GRSP and T-GRSP (Figure 4). In contrast, the PH application didn’t significantly increase the contents of EE-GRSP and T-GRSP. Compared to Ri inoculation, the contents of EE-GRSP and T-GRSP were increased in PH application and Ri inoculation treatment. However, this difference did not reach statistical significance.

### 3.6. Effects of PHs and Ri on AQPs and SOSs Expressions in Root of Citrus Exposed to Salt Stress

A total of eight *PIPs* (*PIP1*, *PIP3*, *PIP5*, *PIP6*, *PIP7*, *PIP8*, *PIP9*, and *PIP10*) were up-regulated by PH application. Six *PIPs* (*PIP1, PIP3, PIP6, PIP7, PIP8,* and *PIP9*) were up-regulated, and two *PIPs* (*PIP2 and PIP5*) were down-regulated by Ri inoculation under salt stress, respectively. Additionally, RP treatment increased the expression of *PIP3* and decreased the expression of *PIP2*, *PIP4*, *PIP5*, *PIP9, and PIP10* (Figure 5A). Except for *TIP5*, *TIP6,* and *TIP7*, most *TIPs* were down-regulated by PH application and Ri inoculation (Figure 5B). Furthermore, PH application and Ri inoculation up-regulated *SOS1* expression under salt stress. However, the expression level of *SOS1* in RP treatment was lower than in the individual treatment but still significantly higher than in control. The expression levels of *SOS2* in the three treatments were significantly lower than control. Three treatments up-regulated *SOS3* expression under salt stress, while the expression level of *SOS3* in Ri and RP treatment was lower than PHs (Figure 5C).

## 4. Discussion

In recent years, salt injury has been the main restriction factor for stable yield and quality improvement of citrus in facilities, especially in whole-year rain shelter cultivation [40]. It is also a significant problem in global agriculture [10]. Notably, the film blocks the leaching of rainwater, and the salt in the surface soil cannot infiltrate into the deeper soil. Additionally, the higher temperature and evaporation in the greenhouse led to the accumulation of salt ions on the soil surface. Excessive fertilization and unreasonable management techniques also lead to excess salt ions in the soil. Citrus is susceptible to salt stress [2]. Consequently, more than 0.2% NaCl in the soil inhibits the growth of citrus roots [41]. Breeding salt-tolerant varieties and rootstocks is a vital way to improve citrus’s salt tolerance, but the citrus breeding cycle lasts for decades [42]. Thus, the research and development of new salt-tolerant cultivation technology is an effective way to sustain citrus cultivation.

### 4.1. PH Application Enhanced the Tolerance of Citrus to Salt Stress by Improving Water Utilization and Sodium Effluxion

PHs are biostimulants rich in polypeptides, oligopeptides, and other active substances, providing essential nutrients for plants and stimulating crop growth to improve tolerance to abiotic stress [43]. In this investigation, PH application promoted citrus growth under salt stress. The aboveground biomass, lateral roots number, and total root length of PH application were significantly higher than the control. These results indicated that PHs effectively alleviated the inhibition effect of salt stress damage on citrus. Likewise, the abundant amino acids in PHs may enhance stress resistance and promote growth. Numerous studies showed that PHs had a similar effect on crops. For example, PHs treatments in corn increased the coleoptile elongation rate, similar to the phenotype of indole-3-acetic acid treatment [44]. Additionally, excessive salt in the soil inhibited the absorption of water and mineral nutrients in crop roots [45]. It is known that *AQPs* are the crucial pathway for plants and microorganisms to transport water and nutrients [11]. Overexpression of *GhPIP2;7* and *GhTIP2;1* enhanced salt and drought tolerance [46]. In this research, eight *PIPs* were up-regulated by PH application, suggesting that PHs might enhance the water utilization of citrus under salt stress via regulating the expression of *PIPs*. Importantly, excess sodium and chloride are toxic to citrus roots [33]. Effluxion of sodium ions by Na^+^/H^+^ antiporter is an effective strategy for plants to cope with salt damage, and *SOSs* are key genes [9,47,48] in regulating sodium effluxion. We found that the expression of *SOS1* and *SOS3* was significantly up-regulated by PH application under salt stress. These results indicate that PHs promote the effluxion of salt ions from roots via up-regulating expression of key genes in the SOS pathway.

### 4.2. AMF Inoculation Enhanced the Tolerance of Citrus to Salt Stress by Improving Water Utilization and Sodium Effluxion

AMF is a fungus that forms a mycorrhizal structure in symbiosis with plants. Plants provide carbohydrates to AMF as an energy substance, and AMF helps plants absorb water and mineral elements [49]. A large number of studies have shown that AMF symbiosis with plants improved resistance to abiotic stress, including drought [50], salt [51], nutrient deficiency [52], and heavy metal stress [53]. In this research, Ri inoculation promoted citrus growth under salt stress. The biomass of aboveground and underground parts of Ri inoculation was significantly higher than that of control. The nitrogen and phosphorus contents of leaves in the Ri treatment were significantly higher than in the control. These results indicated that AMF effectively alleviated the inhibition effect of salt damage on citrus growth. GRSP, which is Glycoprotein secreted by AMF, significantly contributes to soil particle aggregation and carbon sequestration. GRSP is also involved in inducible stress responses in AMF for salinity [23]. The higher contents of EE-GRSP and T-GRSP were found in Ri inoculation treatment, which was conducive to modifying citrus rhizospheric soil aggregates in response to salt stress. Excessive sodium ions reduce the soil’s water potential and hinder the roots’ absorption, resulting in physiological drought [45]. Thus, the developed mycelium of AMF might improve the water absorption of citrus under salt stress. Notably, *AQPs* are key in regulating plant water absorption, transport, and distribution [11]. The AQP gene expression was induced by salt stress in various crops [54,55,56]. Overexpression of the *AQP* gene enhanced salt tolerance of wheat [57], soybean [13], and banana [58]. *PIP* is also involved in the loss of water from plant cells and up-regulating expression of *PIP* may decrease salt and drought stress tolerance [59]. For example, ectopic overexpression of *GsPIP2;1* increases sensitivity to salt and dehydration in transgenic *Arabidopsis thaliana* [60]. Meanwhile, many studies have shown that *AQP* expression is regulated by AMF [25,29,30]. In this research, six *PIPs* and three *TIPs* were up-regulated by Ri inoculation under salt stress. In trifoliate orange, fourteen AQPs were dramatically induced by AMF inoculation under salt stress [30]. This result shows that AMF can increase the expression of *AQP* genes under salt stress, which improves the membrane’s water permeability and stimulates the host’s water transport. Moreover, excess sodium and chloride ions inhibit citrus root growth and nutrient absorption [61]. Thus, effluxion of sodium by Na^+^/H^+^ antiporter is an effective strategy for plants to cope with salt damage [9,47,48]. Previous research suggests that the expression of *SOSs* is regulated by AMF [25,26]. In this study, *SOS1* and *SOS3* were significantly induced by Ri inoculation. The sodium content of leaves in the Ri treatment was also significantly lower than the control. This may be explained by the fact that AMF inoculation can enhance the salt tolerance of citrus.

### 4.3. Interaction of PHs and AMF Enhanced the Tolerance of Citrus to Salt Stress

Although the application of PHs and AMF promoted plant growth and stress resistance has been confirmed in numerous crops, the effect of the combined application of PHs and AMF remains unclear. Previous studies showed that different classes of biostimulants, like PHs, humic substances, seaweed extracts, and microorganisms, synergistically improved crop performance, yield stability, and abiotic stress resistance [62]. This research observed an interaction between factors (PHs × Ri) for stem diameter, leaf number, and leaf mineral element content. Meanwhile, the PH application prompted the colonization of AMF on citrus root. Compared to the individual Ri treatment, the total colonization rate of RP increased by 21%. PHs is rich in free amino acids, polypeptides, and oligopeptides [63], providing sufficient nutrients for the growth of plants and rhizosphere microorganisms, including AMF. Meanwhile, the application of PHs can effectively improve the physicochemical properties of soil, creating a suitable rhizosphere microenvironment for the growth of soil microorganisms and then increasing the biomass of soil microorganisms. The study on lettuce also found that PHs significantly increased the colonization rate of AMF under salt and alkali stress [16]. Chicken feather hydrolysate promoted wheat root colonization by AMF under low P supply [64]. Moreover, the hyphae colonization rate, total colonization rate, and hyphal density in RP treatment were significantly higher than in Ri inoculation. These results indicate that, under salt stress, PHs promoted colonization of AMF on citrus roots, thus forming more mycelial structures and secreting more GRSP. Furthermore, the expression of most *PIPs* and *SOSs* in RP treatment was significantly lower than in PHs and Ri treatment. This may be because PHs promote the colonization of AMF in citrus roots, and the stronger mycelial system enhances the absorption of water and nutrients in citrus under salt stress and the efflux of sodium [65]. Therefore, the interaction between PHs and AMF and the symbiosis between AMF and citrus enhanced the tolerance of citrus to salt stress.

## 5. Conclusions

This study’s PH application and AMF inoculation promoted citrus growth, nutrient absorption, and sodium effluxion under salt stress. Notably, PHs promote the colonization of AMF on citrus. PHs and AMF enhanced citrus salt tolerance by improving water and nutrient absorption and sodium ions’ effluxion via up-regulating *PIP* and *SOS* expression. Furthermore, PHs and AMF interacted positively on citrus under salt stress. In crop cultivation, PHs and AMF can be used as environmentally friendly biostimulants to enhance resistance to environmental stress. The mixed application of PHs and AMF had a better effect. However, the interaction and molecular regulatory mechanism between PHs and AMF has yet to be further investigated.

## Figures and Tables

**Figure 1 jof-09-00983-f001:**
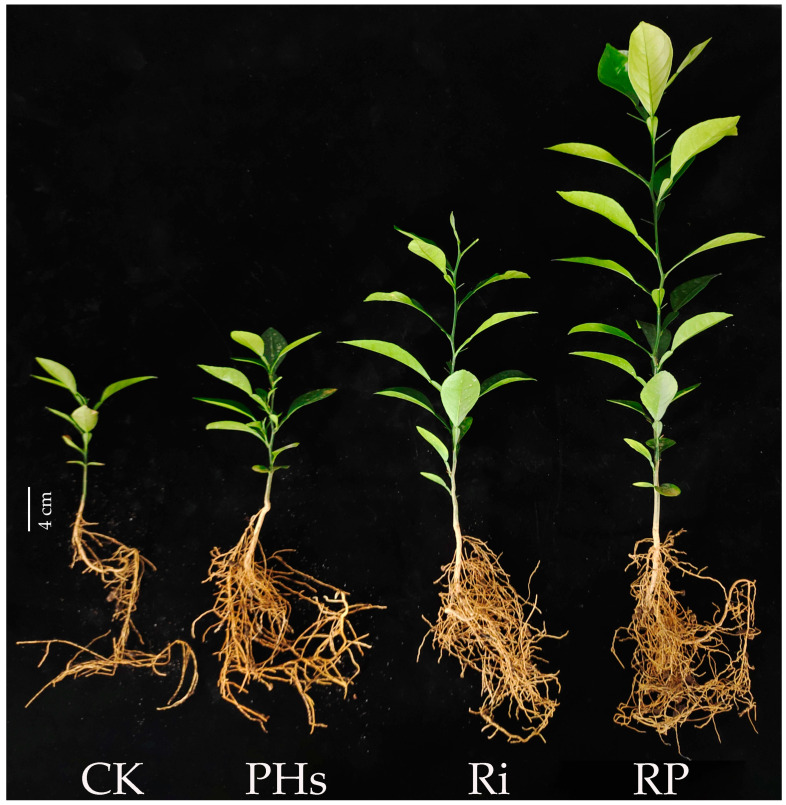
Photos of the effect of PH application and Ri inoculation on citrus plant growth under salt stress. The white scale represents 4 cm. CK refers to control, PHs refers to protein hydrolysate application, Ri refers to *Rhizophagus intraradices* inoculation, and RP refers to the mixed treatment of protein hydrolysate application and *Rhizophagus intraradices* inoculation.

**Figure 2 jof-09-00983-f002:**
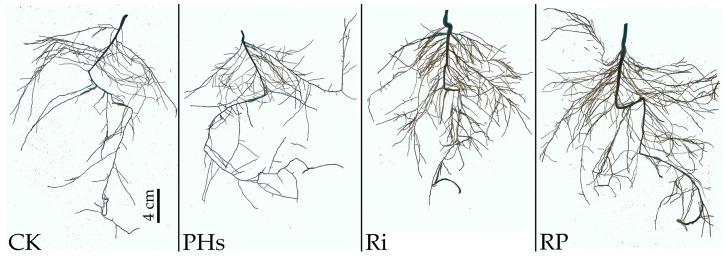
Photos of PH application and Ri inoculation effect on citrus root structure under salt stress. The black scale represents 4 cm. CK refers to control, PHs refers to protein hydrolysate application, Ri refers to *Rhizophagus intraradices* inoculation, and RP refers to mixed treatment of protein hydrolysate application and *Rhizophagus intraradices* inoculation.

**Figure 3 jof-09-00983-f003:**
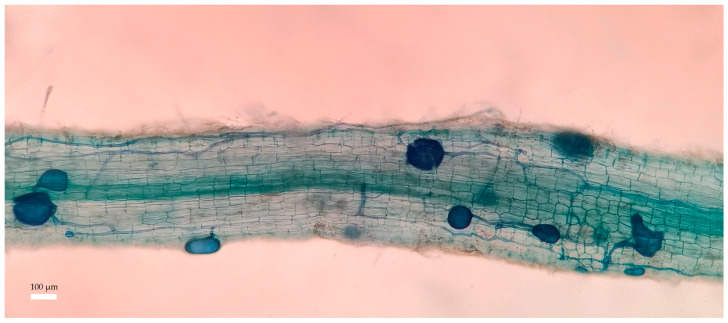
Photo of root colonization of citrus plants inoculated by Ri. The white scale represents 100 μm.

**Figure 4 jof-09-00983-f004:**
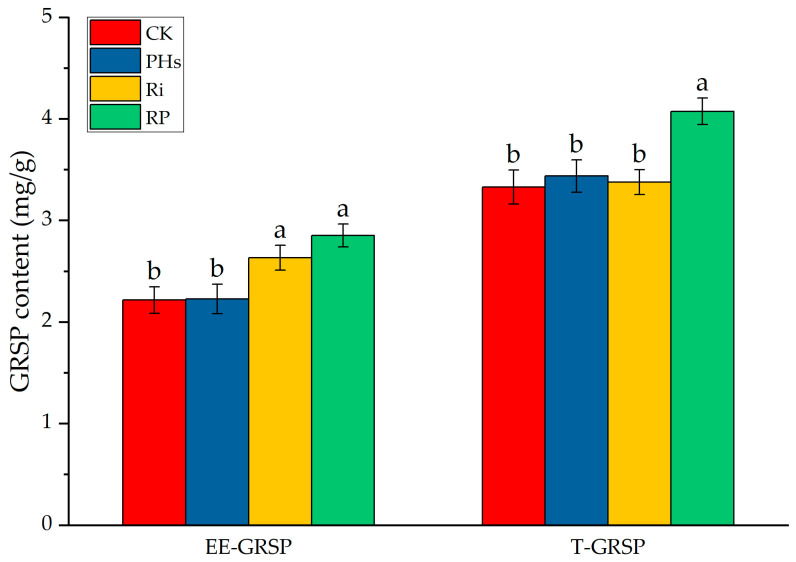
Changes in rhizospheric GRSP contents of citrus by PH application and Ri inoculation under salt stress. CK refers to control, PHs refers to protein hydrolysate application, Ri refers to *Rhizophagus intraradices* inoculation, and RP refers to the mixed treatment of protein hydrolysate application and *Rhizophagus intraradices* inoculation. Different lower-case letters on the bar indicate significant differences (*p* ≤ 0.05).

**Figure 5 jof-09-00983-f005:**
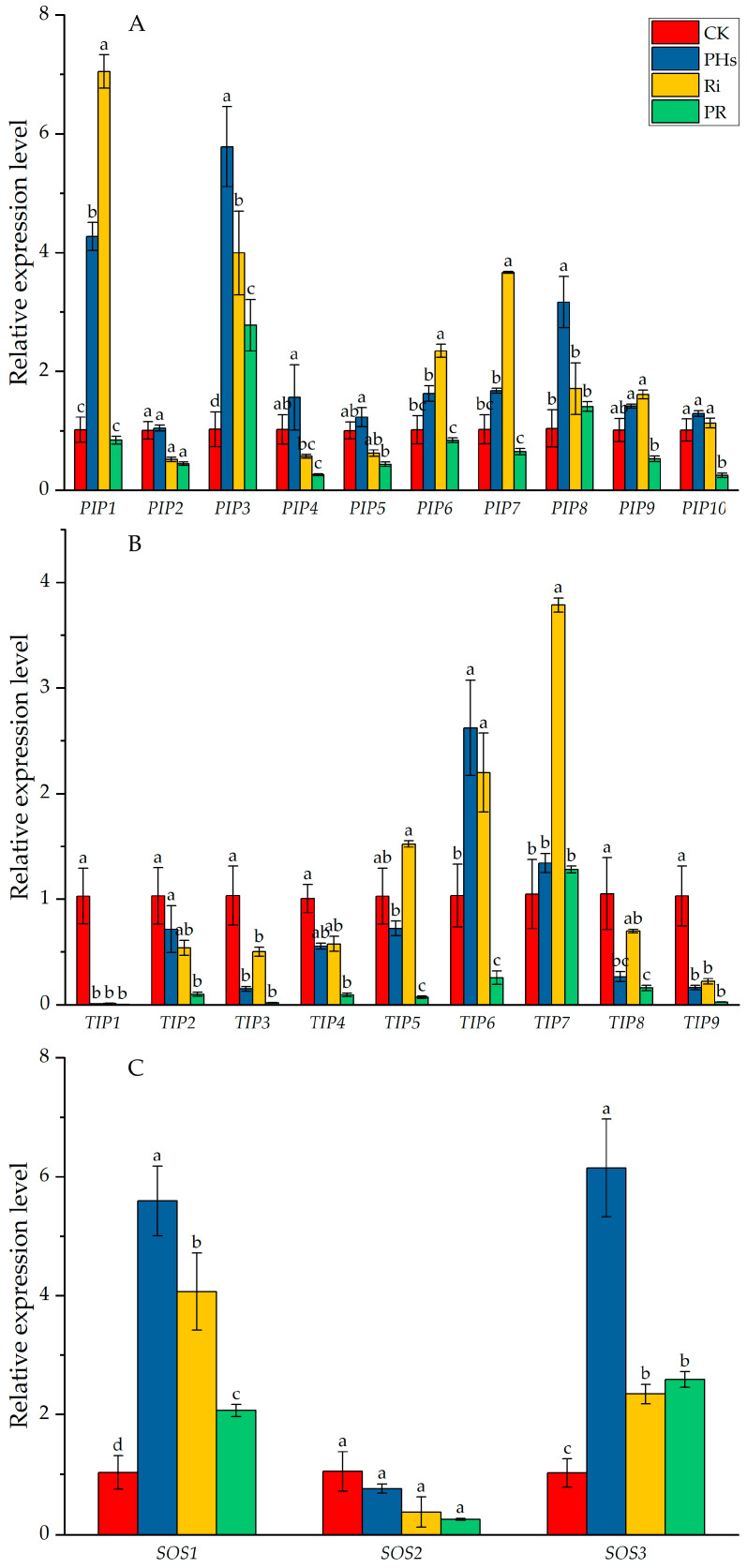
Changes in root *PIPs* (**A**), *TIPs* (**B**), and *SOSs* (**C**) expressions of citrus plants by PH application and Ri inoculation under salt stress. Data (means ± SD, *n* = 3) followed by different letters on the bar indicate significant differences among treatments at the 0.05 level. CK refers to control, PHs refers to protein hydrolysate application, Ri refers to *Rhizophagus intraradices* inoculation, and RP refers to the mixed treatment of protein hydrolysate application and *Rhizophagus intraradices* inoculation. Different lower-case letters on the bar indicate significant differences (*p* < 0.05).

**Table 1 jof-09-00983-t001:** Changes in plant growth parameters of citrus by PH application and Ri inoculation under salt stress.

Treatments	Plant Height (cm)	Stem Diameter (mm)	Leaf Number	Shoot Dry Weight (g)
CK	8.7 ± 0.44 c	1.6 ± 0.12 c	6.8 ± 0.32 c	0.61 ± 0.06 c
PHs	12.2 ± 0.27 b	2.2 ± 0.06 b	10.8 ± 0.19 b	1.08 ± 0.07 b
Ri	20.1 ± 3.33 a	2.9 ± 0.10 a	17.6 ± 1.50 a	2.88 ± 0.43 a
PHs + Ri	23.9 ± 3.18 a	2.8 ± 0.38 a	18.3 ± 1.56 a	3.54 ± 0.59 a
Statistical significance				
PHs	*	*	***	*
Ri	***	***	***	***
PHs×Ri	ns	**	*	ns

All data are expressed as average per plant. ns, *, **, and ***: non-significant or significant at *p* ≤ 0.05, *p* ≤ 0.01, and *p* ≤ 0.001 respectively. Different letters within each parameter indicate statistically significant differences in the same factor according to *t*-test or one-way ANOVA by Tukey’s HSD test at *p* ≤ 0.05. Number of biological replicates (*n* ≥ 5). CK refers to control, PHs refers to protein hydrolysate application, Ri refers to *Rhizophagus intraradices* inoculation, RP refers to the mixed treatment of protein hydrolysate application, and *Rhizophagus intraradices* inoculation.

**Table 2 jof-09-00983-t002:** Changes in mineral content of citrus leaves under salt stress by PH application and Ri inoculation.

Treatments	N (g/kg DW)	P (g/kg DW)	K (g/kg DW)	Ca (g/kg DW)	Mg (g/kg DW)	Na (g/kg DW)
CK	21.86 ± 1.12 b	1.35 ± 0.03 c	13.43 ± 0.55 b	35.52 ± 1.01 c	4.19 ± 0.58 b	5.83 ± 0.56 a
PHs	29.48 ± 1.42 a	2.16 ± 0.15 b	16.07 ± 0.31 a	41.91 ± 0.87 b	6.50 ± 0.36 a	1.45 ± 0.14 b
Ri	27.59 ± 1.02 a	2.26 ± 0.13 b	8.42 ± 0.49 c	45.54 ± 1.34 a	3.61 ± 0.42 b	1.22 ± 0.19 bc
PHs + Ri	28.97 ± 0.99 a	3.11 ± 0.19 a	7.72 ± 0.60 c	43.29 ± 1.11 ab	3.44 ± 0.27 b	1.18 ± 0.22 c
Statistical significance						
PHs	***	***	*	***	*	***
Ri	**	***	***	***	***	***
PHs×Ri	**	ns	***	***	**	***

All data are expressed as average per plant. ns, *, **, and ***: non-significant or significant at *p* ≤ 0.05, *p* ≤ 0.01, and *p* ≤ 0.001 respectively. Different letters within each parameter indicate statistically significant differences in the same factor according to the *t*-test or one-way ANOVA by Tukey’s HSD test at *p* ≤ 0.05. Number of biological replicates (*n* ≥ 5). CK refers to control, PHs refers to protein hydrolysate application, Ri refers to *Rhizophagus intraradices* inoculation, and RP refers to the mixed treatment of protein hydrolysate application and *Rhizophagus intraradices* inoculation.

**Table 3 jof-09-00983-t003:** Change in root structure parameters of citrus after PH application and Ri inoculation under salt stress.

Treatments	Lateral Roots Number	Root Volume (mL)	Total Root Length (cm)	Root Projection Area (cm^2^)	Root Surface Area (cm^2^)	Root Activity (mg/g∙h)
CK	97.8 ± 6.71 d	1.0 ± 0.14 b	306.88 ± 16.63 d	2.61 ± 0.21 b	8.19 ± 0.65 b	0.17 ± 0.02 d
PHs	118.5 ± 5.99 c	1.3 ± 0.19 b	357.10 ± 16.97 c	3.04 ± 0.46 b	9.01 ± 1.86 b	0.26 ± 0.02 c
Ri	183.9 ± 15.66 b	3.0 ± 0.41 a	675.88 ± 33.56 b	5.86 ± 0.67 a	18.41 ± 2.11 a	0.33 ± 0.02 b
PHs + Ri	214.8 ± 15.99 a	3.5 ± 0.45 a	765.17 ± 46.11 a	6.34 ± 1.30 a	19.92 ± 4.08 a	0.40 ± 0.01 a
Statistical significance						
PHs	**	ns	*	ns	ns	***
Ri	***	***	***	***	***	***
PHs×Ri	ns	ns	ns	ns	ns	ns

All data are expressed as average per plant. ns, *, **, and ***: non-significant or significant at *p* ≤ 0.05, *p* ≤ 0.01, and *p* ≤ 0.001 respectively. Different letters within each parameter indicate statistically significant differences in the same factor according to *t*-test or one-way ANOVA by Tukey’s HSD test at *p* ≤ 0.05. Number of biological replicates (*n* ≥ 5). CK refers to control, PHs refers to protein hydrolysate application, Ri refers to *Rhizophagus intraradices* inoculation, and RP refers to the mixed treatment of protein hydrolysate application and *Rhizophagus intraradices* inoculation.

**Table 4 jof-09-00983-t004:** Changes in mycorrhizal colonization parameters of citrus plants under salt stress.

Treatments	Hyphae Colonization Rate (%)	Arbuscule Colonization Rate (%)	Vesicle Colonization Rate (%)	Total Colonization Rate (%)	Spore Density (per/10 g)	Hyphal Density (cm/g)
CK	/	/	/	/	/	/
PHs	/	/	/	/	/	/
Ri	32.69 ± 3.91 b	13.26 ± 2.99 a	5.55 ± 2.86 a	35.18 ± 2.97 b	59.75 ± 13.35 a	19.02 ± 1.13 b
PHs + Ri	39.69 ± 2.61 a	14.80 ± 1.74 a	6.53 ± 2.87 a	42.52 ± 2.50 a	60.25 ± 5.80 a	24.40 ± 2.01 a

All data are expressed as average per plant. Different letters within each parameter indicate statistically significant differences in the same factor according to *t*-test or one-way ANOVA by Tukey’s HSD test at *p* ≤ 0.05. Number of biological replicates (*n* ≥ 5). CK refers to control, PHs refers to protein hydrolysate application, Ri refers to Rhizophagus intraradices inoculation, and RP refers to the mixed treatment of protein hydrolysate application and Rhizophagus intraradices inoculation.

## Data Availability

All the data supporting the findings of this study are included in this article.

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
