# Peer review of "Effects of Interaction of Protein Hydrolysate and Arbuscular Mycorrhizal Fungi Effects on Citrus Growth and Expressions of Stress-Responsive Genes (Aquaporins and SOSs) under Salt Stress"

_jof, 2023, doi:10.3390/jof9100983_

Round 1

Reviewer 1 Report

This publication presents interesting results of the influence of protein hydrolysate and AMF application on citrus growth and aquaporins and SOSs genes expression under salt stress. The adopted approach in this study was very interesting since both biostimulants showed a promising asset to alleviate the deleterious effects of salinity on crops with great interest.

The manuscript was well introduced, and the authors adopted very convincing methods with a consistent discussion of the different obtained results. However, the manuscript needs substantial revisions and English review to be suitable for publication in Journal of Fungi.

General comments

- Comment 1: The English of this manuscript needs substantial improvements

- Comment 2: The description of results should be improved.

- Comment 3: Discussion section should be more developed.

Other comments

Abstract

1. L11: please change “Hydrolysate stimulates” to “Protein hydrolysates”.

2. L13: “but the mechanism is still unclear”, which mechanism, please specify.

3. L16 and L18: please delete “the”.

4. Please state which treatment was the most performant at the end of the abstract.

5. Please add “protein hydrolysates” as a keyword.

Introduction

6. L29: “7 million tons“ which year and please provide FAO reference.

7. L42: please remove repeated “in”.

8. L58: please change “(GRSP) a special” to “(GRSP) is a special“.

9. L59: please change “contributes significantly” to “significantly contributes”.

10. L81: please change “plant biostimulants” to “PHs”.

M&M

11. L88 and L89: please add a space between the value and the unit. Please check throughout the manuscript.

12. L89: please change “PH6.07” to “pH 6.07”.

13. L91: please provide the country of the manufacturer (Italpollina).

14. L94: please change “An” to The”.

15. L99: please change “conducted” to “applied”.

16. L99-100: for the control treatment no biostimulants should be applied? You stated that no AMF inoculation was applied for CK, what about PHs?

17. L103: please change “detected” to “determined”.

18. L105 and 107: please provide the details of equipment used.

19. L114: please change “were” to “was”.

20. L125: please change “use” to “used”.

21. L126-129: please provide more information about the statistical analysis (One-way ANOVA).

Results

22. L132: please change “could promote” to “promoted” and perform the same change for the same situation.

23. L134: please change “with” to “to” and perform the same change for the same situation.

24. L136: please add “respectively” after “77.05%“ and perform the same change for the same situation.

25. Please add a scale bar in Figures 1, 2 and 3.

26. L143-144: please change the title to “Photos of the effect of PHs application and Ri inoculation on citrus plants growth under salt stress.

27. Table 2: Please add the missing footnote.

28. Figure 2: please change the title to “Photos of the effect of PHs application and Ri inoculation on citrus root structure under salt stress.

29. Figure 3: please change the title to “Photo of root colonization of citrus plants inoculated by Ri”.

30. L190: please change “T-GRSP, PHs” to “T-GRSP while PHs”.

31. Figure 4: please add the unit to the Y axis title.

32. L195: please change “Rhizospheric GRSP” to “Changes in rhizospheric GRSP”.

33. L199: please change “A total eight” to “A total of eight”.

34. L202: please remove “While”.

35. L207: please change “three treatments” to “the three treatments”.

Discussion

36. This section should be more developed especially the subsection 4.3. with more emphasize on the interaction between both biostimulants to mitigate salt stress effects on citrus.

37. L217-218: please place “in recent years” at the beginning of the sentence.

38. L219: please change “world” to “worldwide”.

39. L234: please change “was” to “were”.

40. L238: please change “in other crops” to “in other crops”.

41. L254: please change “fatty acids” to “carbohydrates”.

42. L262: please change “ contributes significantly” to “significantly contributes”

43. L275-277: please provides the references of these studies.

44. L279: please change “these” to “this”.

45. L299-301: What are the mechanism?

46. L302: please change “This may be that PHs promotes” to “This may be explained by the fact that PHs promotes”

Conclusion

47. A comprehensive conclusion covering all aspects should be given highlighting the most effective treatment to alleviate salt stress on citrus.

The english of the manuscript needs substantial editing.

Author Response

This publication presents interesting results of the influence of protein hydrolysate and AMF application on citrus growth and aquaporins and SOSs genes expression under salt stress. The adopted approach in this study was very interesting since both biostimulants showed a promising asset to alleviate the deleterious effects of salinity on crops with great interest.

The manuscript was well introduced, and the authors adopted very convincing methods with a consistent discussion of the different obtained results. However, the manuscript needs substantial revisions and English review to be suitable for publication in Journal of Fungi.

General comments

- Comment 1: The English of this manuscript needs substantial improvements

Answer: Thank you for your comments. We carefully checked and revised the grammatical mistake according your advice, and used AiMi Editor (www.aimieditor.com.cn) for English improvement.

- Comment 2: The description of results should be improved.

Answer: Thank you for your comments. We revised the description of the results in the revision.

- Comment 3: Discussion section should be more developed.

Answer: Thank you for your comments. We supplemented the content of the discussion in the revision

Other comments

Answer: Thank you very much for your professional comments and careful revision. We have carefully corrected the errors you pointed out in the revised version.

Abstract

  1. L11: please change “Hydrolysate stimulates” to “Protein hydrolysates”.
  2. L13: “but the mechanism is still unclear”, which mechanism, please specify.
  3. L16 and L18: please delete “the”.
  4. Please state which treatment was the most performant at the end of the abstract.
  5. Please add “protein hydrolysates” as a keyword.

Answer: Thank you for your comments. We revised the abstract in the revision.

Introduction

  1. L29: “7 million tons“ which year and please provide FAO reference.

Answer: Thank you for your comments. The data from FAO in 2021.

  1. L42: please remove repeated “in”.
  2. L58: please change “(GRSP) a special” to “(GRSP) is a special“.
  3. L59: please change “contributes significantly” to “significantly contributes”.
  4. L81: please change “plant biostimulants” to “PHs”.

Answer: Thank you for your comments. We revised the introduction in the revision.

M&M

  1. L88 and L89: please add a space between the value and the unit. Please check throughout the manuscript.
  2. L89: please change “PH6.07” to “pH 6.07”.
  3. L91: please provide the country of the manufacturer (Italpollina).
  4. L94: please change “An” to The”.
  5. L99: please change “conducted” to “applied”.
  6. L99-100: for the control treatment no biostimulants should be applied? You stated that no AMF inoculation was applied for CK, what about PHs?

Answer: PHs treatment followed the product instructions: PHs solution (0.2%) was irrigated 10 days after seedling transplantation, once a week for 10 weeks. Ri treatment: 20 g Ri (1 g contains 20 spores) was inoculated in one pot at seedling transplantation. RP treatment: PHs and Ri were applied together. CK: distilled water instead of PHs irrigation and non-inoculation Ri were conducted.

  1. L103: please change “detected” to “determined”.
  2. L105 and 107: please provide the details of equipment used.
  3. L114: please change “were” to “was”.
  4. L125: please change “use” to “used”.
  5. L126-129: please provide more information about the statistical analysis (One-way ANOVA).

Answer: Thank you for your comments. We revised the materials and methods in the revision.

Results

  1. L132: please change “could promote” to “promoted” and perform the same change for the same situation.
  2. L134: please change “with” to “to” and perform the same change for the same situation.
  3. L136: please add “respectively” after “77.05%“ and perform the same change for the same situation.
  4. Please add a scale bar in Figures 1, 2 and 3.
  5. L143-144: please change the title to “Photos of the effect of PHs application and Ri inoculation on citrus plants growth under salt stress.
  6. Table 2: Please add the missing footnote.
  7. Figure 2: please change the title to “Photos of the effect of PHs application and Ri inoculation on citrus root structure under salt stress.
  8. Figure 3: please change the title to “Photo of root colonization of citrus plants inoculated by Ri”.
  9. L190: please change “T-GRSP, PHs” to “T-GRSP while PHs”.
  10. Figure 4: please add the unit to the Y axis title.
  11. L195: please change “Rhizospheric GRSP” to “Changes in rhizospheric GRSP”.
  12. L199: please change “A total eight” to “A total of eight”.
  13. L202: please remove “While”.
  14. L207: please change “three treatments” to “the three treatments”.

   Answer: Thank you for your comments. We revised the results in the revision.

Discussion

  1. This section should be more developed especially the subsection 4.3. with more emphasize on the interaction between both biostimulants to mitigate salt stress effects on citrus.

Answer: Thank you for your comments. We add more content on the interaction between both biostimulants to mitigate salt stress effects on citrus.

  1. L217-218: please place “in recent years” at the beginning of the sentence.
  2. L219: please change “world” to “worldwide”.
  3. L234: please change “was” to “were”.
  4. L238: please change “in other crops” to “in other crops”.
  5. L254: please change “fatty acids” to “carbohydrates”.
  6. L262: please change “ contributes significantly” to “significantly contributes”
  7. L275-277: please provides the references of these studies.
  8. L279: please change “these” to “this”.
  9. L299-301: What are the mechanism?
  10. L302: please change “This may be that PHs promotes” to “This may be explained by the fact that PHs promotes”

 Answer: Thank you for your comments. We revised the grammar mistakes in the revision.

Conclusion

  1. A comprehensive conclusion covering all aspects should be given highlighting the most effective treatment to alleviate salt stress on citrus.

Answer: Thank you for your comments. We revised the conclusion in the revision.

Reviewer 2 Report

1.      In my opinion, the title should be “Effects of Interaction of Protein Hydrolysate and Arbuscular Mycorrhizal Fungi on Citrus Growth and Expressions of Stress-Responsive Genes (Aquaporins and SOSs) under Salt Stress”

2.      In line 11, should it be “Protein hydrolysate (PH)” instead?

3.      In line 15, what type of citrus was it? Could you make it clearer?

4.      The methodology in the abstract is insufficient and not clear. It should be a bit more detailed. Moreover, please define the saline stress, at what concentration of salt was it?

5.      Aquaporins and SOSs genes should be clearly defined in the abstract

6.      PHS in line 20 should be uniformed in line 11

7.      A kind of citrus should be introduced in background sentence

8.      Further application or research based on the current study should be mentioned at the end of the abstract.

9.      In line 29, “(FAO)”? Is it a citation?

10.  In lines 32-35, are there any literatures for these arguments?

11.  In line 87, the “salinized soil” here is too ambiguous, please describe the soil more clearly, along with the exact salinity.

12.  Organic in line 89, it is an organic matter? line 94, please check the capital

13.  In my opinion, the treatments in the current study lack a variation of different levels of applying the AMF and the PHs. In another word, the number of treatments is inadequate. Moreover, the preparation of the fungal inoculation is insufficient. Furthermore, the amount of AMF and PHs used in the current study lacks a strong foundation or basis. For example, why 0.2% of PHs was irrigated 10 days after plantation, and once a week? and how do you know 20g of Ri was enough?

14.  In line 100, “clear water”? this term should not be used. Distilled water or sterilized water or autoclaved water are more preferable.

15.  The thermal cycle is missing.

16.  TTF in table 3 should have a full form; and please definition of other abbreviations

17.  It would be more preferable if the discussion linked the functions of the genes and the results of the current study together. For example, the application of PHs increased the expression of SOS1 and SOS3 genes, so what can happen? The same should be followed by the rest of the discussion.

18.  The manuscript should be carefully revised, because the language is inadequate in the current version along with many typos. Perhaps, a professional editing service or a native speaker would be a choice as long as the manuscript after resubmission shows a better quality and readability.

19.  As similar to the comment 6, a further application or research based on the current study should be mentioned in the conclusion.

Moderate editing of English language required

Author Response

  1. In my opinion, the title should be “Effects of Interaction of Protein Hydrolysate and Arbuscular Mycorrhizal Fungi on Citrus Growth and Expressions of Stress-Responsive Genes (Aquaporinsand SOSs) under Salt Stress”

Answer: Thank you for your comments. We revised it.

  1. In line 11, should it be “Protein hydrolysate (PH)” instead?

Answer: Thank you for your comments. We revised it.

  1. In line 15, what type of citrus was it? Could you make it clearer?

Answer: Goutoucheng (Citrus aurantium) is a local citrus rootstock resistant to salt.

  1. The methodology in the abstract is insufficient and not clear. It should be a bit more detailed. Moreover, please define the saline stress, at what concentration of salt was it?

Answer: Thank you for your comments. We revised it.

  1. Aquaporins andSOSs genes should be clearly defined in the abstract

Answer: Thank you for your comments. We proofed and revised it.

  1. PHS in line 20 should be uniformed in line 11

Answer: We revised it.

  1. A kind of citrus should be introduced in background sentence

Answer: Thank you for your comments. We added it.

  1. Further application or research based on the current study should be mentioned at the end of the abstract.

Answer: Thank you for your advice. We added relevant content at the end of the abstract.

  1. In line 29, “(FAO)”? Is it a citation?

Answer: The data comes from the FAO (https://www.fao.org/faostat/zh/#home) in 2021.

  1. In lines 32-35, are there any literatures for these arguments?

Answer: Thank you for your advice. We added relevant literatures.

  1. In line 87, the “salinized soil” here is too ambiguous, please describe the soil more clearly, along with the exact salinity.

Answer: Thank you for your comments. The soluble salt content of salinized soil is 0.63%. We described it exactly in the revision.

  1. Organic in line 89, it is an organic matter? line 94, please check the capital

Answer: Thank you for your comments. We revised it.

  1. In my opinion, the treatments in the current study lack a variation of different levels of applying the AMF and the PHs. In another word, the number of treatments is inadequate. Moreover, the preparation of the fungal inoculation is insufficient. Furthermore, the amount of AMF and PHs used in the current study lacks a strong foundation or basis. For example, why 0.2% of PHs was irrigated 10 days after plantation, and once a week? and how do you know 20g of Ri was enough?

Answer: Thank you for your comments. The fertilization amount and frequency of PHs were followed the product instructions. The AMF inoculation amount was followed previous research. The spore density of AMF agent in this research is 20 unit/g. The spore number of 20g/pot is about 400. So, it is enough for the experiment. We added exact description in the revision.

  1. In line 100, “clear water”? this term should not be used. Distilled water or sterilized water or autoclaved water are more preferable.

Answer: Thank you for your comments. We used distilled water in the experiment.

  1. The thermal cycle is missing.

Answer: Thank you for your comments. We added it in the revision.

  1. TTF in table 3 should have a full form; and please definition of other abbreviations

Answer: The unit of root activity is incorrect. We revised it in the revision.

  1. It would be more preferable if the discussion linked the functions of the genes and the results of the current study together. For example, the application of PHs increased the expression of SOS1and SOS3 genes, so what can happen? The same should be followed by the rest of the discussion.

Answer: Thank you for your advice. We added relevant content to the discussion.

  1. The manuscript should be carefully revised, because the language is inadequate in the current version along with many typos. Perhaps, a professional editing service or a native speaker would be a choice as long as the manuscript after resubmission shows a better quality and readability.

Answer: Thank you for your advice. We used AiMi Editor (www.aimieditor.com.cn) for grammar improvement.

  1. As similar to the comment 6, a further application or research based on the current study should be mentioned in the conclusion.

Answer: Thank you for your advice. We added relevant content to the conclusion.

Reviewer 3 Report

The authors have analyzed the effects of Protein Hydrolysate (PH) and Arbuscular Mycorrhizal Fungi (AMF) on seeding growth, physiological activities, and the expression of stress responsive genes such as SoSs, TIPs and PIPs were analyzed in Citrus aurantium, a local citrus rootstock resistant to salt stress. For which, authors have performed some series of experiments. The study revealed that, AMF and PHs enhanced the salt tolerance of citrus by ameliorating the nutrient absorption and sodium effluxion by elevating the expression of PIPs and SOSs.

However, the manuscript still needs improvisation. Therefore, I recommend the authors to incorporate the following points in the manuscript for further consideration.

Line 11: Include the word ‘Protein’ before the Hydrolysate stimulates.

Line 29: Authors are advised to provide FAO report link and year.

Line 121: Gene names should be italics.

Please describe more about in the RNA extraction section in the materials and method section. What is RNA concentration? How much concentration was used for cDNA conversion.

qRT-PCR Analysis section needs to be improved and cite the appropriate reference in section 2.3.
Authors must improve the data analysis section. The section should contain the details Whether you have performed the experiments in biological replicates, experimental replicates, or duplicates or not.

All figure description should be self-explanatory instead of just mentioning the title of the figures. Mentioning the CK, PHs, Ri, RP in the figure is necessary.  

Make the figure 4, 5 in color to easily distinguish the differences easily by the readers.

Authors have mentioned Table 4 as Table 3. Correct it. Provide table 4 foot notes for clear understanding.

In figure 5 gene names should be italicized. The figure 5 is not clear. Revise it.

Conclusion section needs to be improved. In addition, write a few lines about future perspectives or hypotheses about the study. It will be useful to the readers for ease of understanding.

Gene names should be in italics. Check and revise the same throughout the manuscript.

Check the space and punctuation errors throughout the manuscript.

Authors have to check the space and punctuation errors throughout the manuscript.

Author Response

The authors have analyzed the effects of Protein Hydrolysate (PH) and Arbuscular Mycorrhizal Fungi (AMF) on seeding growth, physiological activities, and the expression of stress responsive genes such as SoSsTIPs and PIPs were analyzed in Citrus aurantium, a local citrus rootstock resistant to salt stress. For which, authors have performed some series of experiments. The study revealed that, AMF and PHs enhanced the salt tolerance of citrus by ameliorating the nutrient absorption and sodium effluxion by elevating the expression of PIPs and SOSs.

However, the manuscript still needs improvisation. Therefore, I recommend the authors to incorporate the following points in the manuscript for further consideration.

Line 11: Include the word ‘Protein’ before the Hydrolysate stimulates.

Answer: Thank you for your comments. We revised it.

Line 29: Authors are advised to provide FAO report link and year.

Answer: Thank you for your comments. We revised it.

Line 121: Gene names should be italics.

Answer: Thank you for your comments. We revised it.

Please describe more about in the RNA extraction section in the materials and method section. What is RNA concentration? How much concentration was used for cDNA conversion.

Answer: Thank you for your advice. RNA concentration of different treatments ranged from 83 to 400 ng/μL. After the integrity and purity testing, 1 μg of the obtained RNA was reversely tran-scribed into cDNA. We added relevant content in the revision.

qRT-PCR Analysis section needs to be improved and cite the appropriate reference in section 2.3.

Answer: Thank you for your advice. We added relevant content in the revision

Authors must improve the data analysis section. The section should contain the details Whether you have performed the experiments in biological replicates, experimental replicates, or duplicates or not.

Answer: Thank you for your advice. We added relevant content in the revision

All figure description should be self-explanatory instead of just mentioning the title of the figures. Mentioning the CK, PHs, Ri, RP in the figure is necessary.  

Answer: Thank you for your comments. We revised it.

Make the figure 4, 5 in color to easily distinguish the differences easily by the readers.

Authors have mentioned Table 4 as Table 3. Correct it. Provide table 4 foot notes for clear understanding.

Answer: Thank you for your comments. We revised it.

In figure 5 gene names should be italicized. The figure 5 is not clear. Revise it.

Answer: Thank you for your comments. We revised it.

Conclusion section needs to be improved. In addition, write a few lines about future perspectives or hypotheses about the study. It will be useful to the readers for ease of understanding.

Answer: Thank you for your comments. We added future perspectives or hypotheses about the study in the conclusion.

Gene names should be in italics. Check and revise the same throughout the manuscript.

Answer: Thank you for your comments. We revised it throughout the manuscript.

Check the space and punctuation errors throughout the manuscript.

Answer: Thank you for your comments. We revised it throughout the manuscript.

Round 2

Reviewer 1 Report

Please place "The white scale represents 4 cm" in Figure 1 caption not in Table 1 caption.

Author Response

Comments and Suggestions for Authors

Please place "The white scale represents 4 cm" in Figure 1 caption not in Table 1 caption.

Answer: Thanks for your comments. We revised the figure note in Figure 1.

Reviewer 3 Report

The authors have addressed all my comments and suitably incorporated them into the revised manuscript. Now the manuscript looks better than the previous version. Therefore, I recommend this manuscript be accepted for publication.

Before final publication, authors should correct the below mentioned minor comment in the manuscript.

In Line 181: “The white scale represents 4 cm’ I think it should be moved to the figure 1 description.

Author Response

Comments and Suggestions for Authors

In Line 181: “The white scale represents 4 cm’ I think it should be moved to the figure 1 description.

Answer: Thanks for your comments. We revised the figure note in Figure 1.